# Loss Functions for Multiset Prediction

## Abstract

We study the problem of multiset prediction. The goal of multiset prediction is to train a predictor that maps an input to a multiset consisting of multiple items. Unlike existing problems in supervised learning, such as classification, ranking and sequence generation, there is no known order among items in a target multiset, and each item in the multiset may appear more than once, making this problem extremely challenging. In this paper, we propose a novel multiset loss function by viewing this problem from the perspective of sequential decision making. The proposed multiset loss function is empirically evaluated on two families of datasets, one synthetic and the other real, with varying levels of difficulty, against various baseline loss functions including reinforcement learning, sequence, and aggregated distribution matching loss functions. The experiments reveal the effectiveness of the proposed loss function over the others.

## 1 Introduction

A relatively less studied problem in machine learning, particularly supervised learning, is the problem of multiset prediction. The goal of this problem is to learn a mapping from an arbitrary input to a multiset[1] of items. This problem appears in a variety of contexts. For instance, in the context of high-energy physics, one of the important problems in a particle physics data analysis is to count how many physics objects, such as electrons, muons, photons, taus, and jets, are in a collision event (Ehrenfeld et al., 2011). In computer vision, automatic alt-text, such as the one available on Facebook,[2] is a representative example of multiset prediction (Welleck et al., 2017; Lempitsky & Zisserman, 2010).[3]

In multiset prediction, a learner is presented with an arbitrary input and the associated multiset of items. It is assumed that there is no predefined order among the items, and that there are no further annotations containing information about the relationship between the input and each of the items in the multiset. These properties make the problem of multiset prediction unique from other well-studied problems. It is different from sequence prediction, because there is no known order among the items. It is not a ranking problem, since each item may appear more than once. It cannot be transformed into classification, because the number of possible multisets grows exponentially with respect to the maximum multiset size.

In this paper, we view multiset prediction as a sequential decision making process. Under this view, the problem reduces to finding a policy that sequentially predicts one item at a time, while the outcome is still evaluated based on the aggregate multiset of the predicted items. We first propose an oracle policy that assigns non-zero probabilities only to prediction sequences that result exactly in the target, ground-truth multiset given an input. This oracle is optimal in the sense that its prediction never decreases the precision and recall regardless of previous predictions. That is, its decision is optimal in any state (i.e., prediction prefix). We then propose a novel *multiset loss* which minimizes the KL divergence between the oracle policy and a parametrized policy at every point in a decision trajectory of the parametrized policy.

---

[1] A set that allows multiple instances, e.g. $\{x, y, x\}$. See Appendix A for a detailed definition.

[2] https://newsroom.fb.com/news/2016/04/using-artificial-intelligence-to-help-blind-people-see-facebook/

[3] We however note that such a multiset prediction problem in computer vision can also be solved as segmentation, if fine-grained annotation is available. See, e.g., (He et al., 2017).

We compare the proposed multiset loss against an extensive set of baselines. They include a sequential loss with an arbitrary rank function, sequential loss with an input-dependent rank function, and an aggregated distribution matching loss and its one-step variant. We also test policy gradient, as was done by Welleck et al. (2017) recently for multiset prediction. Our evaluation is conducted on two sets of datasets with varying difficulties and properties. According to the experiments, we find that the proposed multiset loss outperforms all the other loss functions.

The paper is structured as follows. We first define multiset prediction at the beginning of Section 2, and compare it to existing problems in supervised learning in 2.1. Then we propose the multiset loss in Section 2.2, followed by alternative baseline losses in Section 3. The multiset loss and baselines are then empirically evaluated in Section 4.

## 2   MULTISET PREDICTION

A multiset prediction problem is a generalization of classification, where a target is not a single class but a multiset of classes. The goal is to find a mapping from an input $x$ to a multiset $\mathcal{Y} = \{y_1, \ldots, y_{|\mathcal{Y}|}\}$, where $y_k \in \mathcal{C}$. Some of the core properties of multiset prediction are

1. the input $x$ is an arbitrary vector.
2. there is no predefined order among the items $y_i$ in the target multiset $\mathcal{Y}$.
3. the size of $\mathcal{Y}$ may vary depending on the input $x$.
4. each item in the class set $\mathcal{C}$ may appear more than once in $\mathcal{Y}$.

Refer to Appendix A for definitions related to multiset prediction.

As is typical in supervised learning, in multiset prediction a model $f_\theta(x)$ is evaluated by forming a test set $\{(x_i, \mathcal{Y}_i)\}_{i=1}^n$ and computing evaluation metrics $m(\cdot)$ that compare the predicted and target multisets, $\frac{1}{n} \sum_{i=1}^n m(\hat{\mathcal{Y}}_i, \mathcal{Y}_i)$, where $\hat{\mathcal{Y}}_i = f_\theta(x_i)$ denotes a predicted multiset. Here, F1 score and exact match (defined in Appendix A), are used as evaluation metrics.

### 2.1   RELATED PROBLEMS IN SUPERVISED LEARNING

Variants of this multiset prediction problem have been extensively studied. However, they differ from our definition of the problem. Here, we go over each variant and discuss how it differs from our definition of multiset prediction.

**Power Multiset Classification**   Perhaps the most naive approach to multiset prediction is to transform the class set $C$ into a set $M(C)$ of all possible multisets. This transformation, or the size of $M(C)$, is not well defined unless some constraints are put in place. If the maximum size of a target multiset is set to $K$, the number of all possible multisets is

$$\sum_{k=1}^{K} \frac{(|C| + k - 1)!}{k!(|C| - 1)!}.$$

With some constant $|C|$, we notice that this grows exponentially in the maximum size of the target multiset.

Once the class set $C$ is transformed, we can train a multi-class classifier $\pi$ that maps an input $x$ to one of the elements in $M(C)$. However, this is infeasible in practice and generally intractable. For instance, for the COCO Medium dataset used later in the experiments (see section 4.1), $M(C)$ has roughly 20 thousand elements while the dataset only contains roughly 40 thousand training examples. For the full MS COCO dataset, $|M(C)|$ is on the order of $10^{49}$, making it infeasible to learn a classifier using this method.

**Ranking**   A ranking problem can be considered as learning a mapping from a pair of input $x$ and one of the items $c \in \mathcal{C}$ to its score $s(x, c)$. All the items in the class set are then sorted according to the score, and this sorted order determines the rank of each item. By taking the top-$K$ items from this sorted list, we can turn this problem of ranking into set prediction. Similarly to multiset

prediction, the input $x$ is arbitrary, and the target is a set without any prespecific order. However, ranking differs from multiset prediction in that it is unable to handle multiple occurrences of a single item in the target set.

**Aggregated Distribution Matching**   Instead of considering the target multiset as an actual multiset, one can convert it into a distribution by computing the frequency of each item from the class set in the target multiset. That is,

$$p(y|x) = \frac{\sum_{y_i \in \mathcal{Y}} I_{y_i = y}}{|\mathcal{Y}|},$$

where $I.$ is an indicator function. Then, we can simply minimize a divergence between this distribution and the predictive distribution from a model. This loss function works only when the conditional distribution $p(y|x)$ substantially differs from the marginal distribution $p(y)$, since the model would resort to a trivial solution of predicting the marginal distribution regardless of the input $x$.

We describe this approach in more detail in Sec. 3.1, and test it against our proposal in the experiments.

**Sequence prediction**   A sequence prediction problem is characterized as finding a mapping from an input $x$ to a sequence of classes $\mathcal{Y} = (y_1, \ldots, y_{|\mathcal{Y}|})$. Representative examples of sequence prediction include machine translation, automatic speech recognition and other tagging problems, such as part-of-speech tagging, in natural language processing. Similarly to multiset prediction, the input $x$ is arbitrary, and an item in the class set $\mathcal{C}$ may appear more than once in the target sequence. It is, however, different from multiset prediction in that there is a clear, predetermined order of items in the target sequence. We detail this sequence prediction approach later in Sec. 3.2.

## 2.2   MULTISET LOSS FUNCTION FOR MULTISET PREDICTION

In this paper, we propose a novel loss function, called *multiset loss*, for the problem of multiset prediction. This loss function is best motivated by treating the multiset prediction problem as a sequential decision making process with a model being considered a policy $\pi$. This policy takes as input the input $x$ and all the previously predicted classes $\hat{y}_{<t}$ at time $t$, and outputs the distribution over the next class to be predicted. That is, $\pi_\theta(y_t | \hat{y}_{<t}, x)$. This policy is parametrized with a set $\theta$ of parameters.

We first define a free label multiset at time $t$ as

**Definition 1** (Free Label Multiset)**.**

$$\mathcal{Y}_t \leftarrow \mathcal{Y}_{t-1} \setminus \{\hat{y}_{t-1}\}$$

$\hat{y}_{t-1}$ is the prediction made by the policy at time $t - 1$.

This free label multiset $\mathcal{Y}_t$ contains all the items that remain to be predicted after $t - 1$ predictions by the policy.

We then construct an oracle policy $\pi_*$. This oracle policy takes as input a sequence of predicted labels $\hat{y}_{<t}$, the input $x$, and the free label multiset with respect to its predictions, $\mathcal{Y}_t = \mathcal{Y} \setminus \{\hat{y}_{<t}\}$. It outputs a distribution whose entire probability (1) is evenly distributed over all the items in the free label multiset $\mathcal{Y}_t$. In other words,

**Definition 2** (Oracle)**.**

$$\pi_*(y_t | \hat{y}_{<t}, x, \mathcal{Y}_t) = \begin{cases} \frac{1}{|\mathcal{Y}_t|}, & \text{if } y_t \in \mathcal{Y}_t \\ 0, & \text{otherwise} \end{cases}$$

An interesting and important property of this oracle is that it is optimal given any prefix $\hat{y}_{<t}$ with respect to both precision and recall. This is intuitively clear by noticing that the oracle policy allows only a correct item to be selected. We call this property the optimality of the oracle.

**Remark 1.** Given an arbitrary prefix $\hat{y}_{<t}$,

$$\text{Prec}(\hat{y}_{<t}, \mathcal{Y}) \leq \text{Prec}(\hat{y}_{<t} \cup \hat{y}, \mathcal{Y}) \text{ and } \text{Rec}(\hat{y}_{<t}, \mathcal{Y}) \leq \text{Rec}(\hat{y}_{<t} \cup \hat{y}, \mathcal{Y}),$$

for any $\hat{y} \sim \pi_*(\hat{y}_{<t}, x, \mathcal{Y}_t)$.

The proof is given in Appendix B. See Appendix A for definitions of precision and recall for multisets.

From the remark above, it follows that the oracle policy is an optimal solution to the problem of multiset prediction in terms of precision and recall.

**Remark 2.**

$$\text{Prec}(\hat{y}_{\leq|\mathcal{Y}|}, \mathcal{Y}) = 1 \text{ and } \text{Rec}(\hat{y}_{\leq|\mathcal{Y}|}, \mathcal{Y}) = 1,$$

for all $\hat{y}_{\leq|\mathcal{Y}|} \sim \pi_*(y_1|x, \mathcal{Y}_1)\pi_*(y_2|y_1, x, \mathcal{Y}_2)\cdots\pi_*(y_{|\mathcal{Y}|}|y_{<|\mathcal{Y}|}, x, \mathcal{Y}_{|\mathcal{Y}|})$.

The proof can be found in Appendix C.

It is trivial to show that sampling from such an oracle policy would never result in an incorrect prediction. That is, this oracle policy assigns zero probability to any sequence of predictions that is not a permutation of the target multiset.

**Remark 3.**

$$\prod_{t=1}^{|\mathcal{Y}|} \pi_*(y_t|y_{<t}, x) = 0, \text{ if multiset}(y_1, \ldots, y_{|\mathcal{Y}|}) \neq \mathcal{Y},$$

where multiset equality refers to exact match, defined in Appendix A. In short, this oracle policy tells us at each time step $t$ which of all the items in the class set $C$ must be selected. This optimality allows us to consider a step-wise loss between a parametrized policy $\pi_\theta$ and the oracle policy $\pi_*$, because the oracle policy provides us with an optimal decision regardless of the quality of the prefix generated so far. We thus propose to minimize the KL divergence from the oracle policy to the parametrized policy at each step separately. This divergence is defined as

$$\text{KL}(\pi_*^t \| \pi_\theta^t) = \underbrace{\mathcal{H}(\pi_*^t)}_{\text{const. w.r.t. } \theta} - \sum_{y_j \in |\mathcal{Y}_t|} \frac{1}{|\mathcal{Y}_t|} \log \pi_\theta(y_j|\hat{y}_{<t}, x), \tag{1}$$

where $\mathcal{Y}_t$ is formed using predictions $\hat{y}_{<t}$ from $\pi_\theta$, and $\mathcal{H}(\pi_*^t)$ is the entropy of the oracle policy at time step $t$. This entropy term can be safely ignored when learning $\pi_\theta$, since it is constant with respect to $\theta$. We define

$$\mathcal{L}^t(x, \mathcal{Y}, \hat{y}_{<t}, \theta) = \text{KL}(\pi_*^t \| \pi_\theta^t) - \mathcal{H}(\pi_*^t) \tag{2}$$

and call it a per-step loss function. We note that it is indeed possible to use another divergence in the place of the KL divergence.

It is intractable to minimize the per-step loss from Eq. (2) for every possible state $(\hat{y}_{<t}, x)$, since the size of the state space grows exponentially with respect to the size of a target multiset. We thus propose here to minimize the per-step loss only for the state, defined as a pair of the input $x$ and the prefix $\hat{y}_{<t}$, visited by the parametrized policy $\pi_\theta$. That is, we generate an entire trajectory $(\hat{y}_1, \ldots, \hat{y}_T)$ by executing the parametrized policy until either all the items in the target multiset have been predicted or the predefined maximum number of steps have passed. Then, we compute the loss function at each time $t$ based on $(x, \hat{y}_{<t})$, for all $t = 1, \ldots, T$. The final loss function is then the sum of all these per-step loss functions.

**Definition 3** (Multiset Loss Function)**.**

$$\mathcal{L}(x, \mathcal{Y}, \theta) = -\sum_{t=1}^{T} \frac{1}{|\mathcal{Y}_t|} \sum_{y_j \in \mathcal{Y}_t} \log \pi_\theta(y_j|\hat{y}_{<t}, x),$$

where $T$ is the smaller of the smallest $t$ for which $\mathcal{Y}_t = \emptyset$ and the predefined maximum number of steps allowed.

Note that as a consequence of Remarks 2 and 3, minimizing the multiset loss function results in maximizing F1 and exact match.

As was shown by Ross et al. (2011), the use of the parametrized policy $\pi_\theta$ instead of the oracle policy $\pi_*$ allows the upper bound on the learned policy's error to be linear with respect to the size of the target multiset. If the oracle policy had been used, the upper bound would have grown quadratically with respect to the size of the target multiset. To confirm this empirically, we test the following three alternative strategies for executing the parametrized policy $\pi_\theta$ in the experiments:

1. Greedy search: $\hat{y}_t = \arg\max_y \log \pi_\theta(y|\hat{y}_{<t}, x)$
2. Stochastic sampling: $\hat{y}_t \sim \pi_\theta(y|\hat{y}_{<t}, x)$
3. Oracle sampling: $\hat{y}_t \sim \pi_*(y|\hat{y}_{<t}, x, \mathcal{Y}_t)$

Once the proposed multiset loss is minimized, we evaluate the learned policy by greedily selecting each item from the policy.

### 2.3 VARIABLE-SIZED TARGET MULTISET

We have defined the proposed loss function for multiset prediction while assuming that the size of the target multiset was known. However, this is a major limitation, and we introduce two different methods for relaxing this constraint.

**Termination Policy** The termination policy $\pi_s$ outputs a stop distribution given the predicted sequence of items $\hat{y}_{<t}$ and the input $x$. Because the size of the target multiset is known during training, we simply train this termination policy in a supervised way using a binary cross-entropy loss. At evaluation time, we simply threshold the predicted stop probability at a predefined threshold (0.5).

**Special Class** An alternative strategy is to introduce a special item to the class set, called $\langle END \rangle$, and add it to the final free label multiset $\mathcal{Y}_{|\mathcal{Y}|+1} = \{\langle END \rangle\}$. Thus, the parametrized policy is trained to predict this special item $\langle END \rangle$ once all the items in the target multiset have been predicted. This is analogous to NLP sequence models which predict an end of sentence token (Sutskever et al., 2014; Bahdanau et al., 2014), and was used in Welleck et al. (2017) to predict variable-sized multisets.

## 3 OTHER LOSS FUNCTIONS

In addition to the proposed multiset loss function, we propose three more loss functions for multiset prediction. They serve as baselines in our experiments later.

### 3.1 AGGREGATED DISTRIBUTION MATCHING

In the case of distribution matching, we consider the target multiset $\mathcal{Y}$ as a set of samples from a single, underlying distribution $q^*$ over the class set $C$. This underlying distribution can be empirically estimated by counting the number of occurrences of each item $c \in C$ in $\mathcal{Y}$. That is,

$$q_*(c|x) = \frac{1}{|\mathcal{Y}|} \sum_{y \in \mathcal{Y}} I_{y=c},$$

where $I$ is the indicator function as before.

Similarly, we can construct an aggregated distribution computed by the parametrized policy $\pi_\theta$. As with the proposed multiset loss in Def. 3, we first execute $\pi_\theta$ to predict a multiset $\hat{\mathcal{Y}}$. This is converted into an aggregated distribution $q_\theta$ in the same way as we turned the target multiset into the oracle aggregate distribution.

Learning is equivalent to minimizing the divergence between these two distributions. In this paper, we test two types of divergences. The first one is from a family of $L_p$ distances defined as

$$\mathcal{L}_{dm}^p(x, \mathcal{Y}, \theta) = \|q_* - q_\theta\|_p,$$

where $q_*$ and $q$ are the vectors representing the corresponding categorical distributions. The other is a usual KL divergence defined earlier in Eq. (1):

$$\mathcal{L}_{dm}^{KL}(x, \mathcal{Y}, \theta) = \sum_{c \in C} q_*(c|x) \log q_\theta(c|x).$$

One major issue with this approach is that minimizing the divergence between the aggregated distributions does not necessarily result in the optimal policy (see the oracle policy in Def. 2.) That is, a policy that minimizes this loss function may assign non-zero probability to an incorrect sequence

of predictions, unlike the oracle policy. This is due to the invariance of the aggregated distribution to the order of predictions. Later when analyzing this loss function, we empirically notice that a learned policy often has a different behaviour from the oracle policy, for instance, reflected by the increasing entropy of the action distribution over time.

**One-Step Variant** We can train an one-step predictor with this aggregate distribution matching criterion, instead of learning a policy $\pi_\theta$. That is, a predictor outputs both a point $q_\theta(\cdot|x)$ in a $|C|$-dimensional simplex and the size $\hat{l}_\theta(x)$ of the target multiset. Then, for each unique item $c \in C$, the number of its occurrences in the predicted multiset $\hat{\mathcal{Y}}$ is

$$\#(c) = \text{round}(q_\theta^c(x) \cdot \hat{l}(x)).$$

The corresponding loss function is then

$$\mathcal{L}_{\text{1-step}}(x, \mathcal{Y}, \theta) = \sum_{c \in C} q_*(c|x) \log q_\theta(c|x) + \lambda(\hat{l}_\theta(x) - |\mathcal{Y}|)^2,$$

where $\lambda > 0$ is a coefficient for balancing the contributions from the two terms.

A major weakness of this one-step variant, compared to the approaches based on sequential decision making, is the lack of modelling dependencies among the items in the predicted multiset. We test this approach in the experiments later and observe this lack of output dependency modelling results in substantially worse prediction accuracy.

## 3.2 SEQUENCE PREDICTION WITH A PREDEFINED ORDER

All the loss functions defined so far have not relied on the availability of an existing order of items in a target multiset. However, by turning the problem of multiset prediction into sequential decision making, minimizing such a loss function is equivalent to capturing an order of items in the target multiset implicitly. Here, we instead describe an approach based on explicitly defining an order in advance. This will serve as a baseline later in the experiments.

We first define a rank function $r$ that maps from one of the unique items in the class set $c \in C$ to a unique integer. That is, $r : C \to \mathbb{Z}$. This function assigns the rank of each item and is used to order items $y_i$ in a target multiset $\mathcal{Y}$. This results in a sequence $\mathcal{S} = (s_1, \ldots, s_{|\mathcal{Y}|})$, where $r(s_i) \geq r(s_j)$ for all $j > i$, and $s_i \in \mathcal{Y}$. With this target sequence $\mathcal{S}$ created from $\mathcal{Y}$ using the rank function $r$, we define a sequence loss function as

$$\mathcal{L}_{\text{seq}}(x, \mathcal{S}, \theta) = -\sum_{t=1}^{|\mathcal{S}|} \log \pi_\theta(s_t|s_{<t}, x).$$

Minimizing this loss function is equivalent to maximizing the conditional log-probability of the sequence $\mathcal{S}$ given $x$.

This sequence loss function has two clear disadvantages. First, it does not take into account the actual behaviour of the policy $\pi_\theta$ (see, e.g., Bengio et al., 2015; Daumé III & Marcu, 2005; Ross et al., 2011). This makes a learned policy potentially vulnerable to cascading error at test time. Second and more importantly, this loss function requires a pre-specified rank function $r$. Because multiset prediction does not come with such a rank function by definition, we must design an arbitrary rank function, and the final performance varies significantly based on the choice. We demonstrate this variation in section 4.3.

**Input-Dependent Rank Function** When the input $x$ has a well-known structure, and an object within the input for each item in the target multiset is annotated, it is possible to devise a rank function per input. A representative example is an image input with bounding box annotations. Here, we present two input-dependent rank functions in such a case.

First, a spatial rank function $r_{\text{spatial}}$ assigns an integer rank to each item in a given target multiset $\mathcal{Y}$ such that

$$r_{\text{spatial}}(y_i|x) < r_{\text{spatial}}(y_j|x), \text{ if } \text{pos}_x(x_i) < \text{pos}_x(x_j) \text{ and } \text{pos}_y(x_i) < \text{pos}_y(x_j),$$

where $x_i$ and $x_j$ are the objects corresponding to the items $y_i$ and $y_j$.

Second, an area rank function $r_{\text{area}}$ decides the rank of each label in a target multiset according to the size of the corresponding object inside the input image:

$$r_{\text{area}}(y_i|x) < r_{\text{area}}(y_j|x), \text{ if area}(x_i) < \text{area}(x_j).$$

The area may be determined based on the size of a bounding box or the number of pixels, depending on the level of annotation.

We test these two image-specific input-dependent rank functions against a random rank function in the experiments.

### 3.3 REINFORCEMENT LEARNING

In (Welleck et al., 2017), an approach based on reinforcement learning was proposed for multiset prediction. Instead of assuming the existence of an oracle policy, this approach solely relies on a reward function $r$ designed specifically for multiset prediction. The reward function is defined as

$$r(\hat{y}_t, \mathcal{Y}_t) = \begin{cases} 1, & \text{if } \hat{y}_t \in \mathcal{Y}_t \\ -1, & \text{otherwise} \end{cases}$$

The goal is then to maximize the sum of rewards over a trajectory of predictions from a parametrized policy $\pi_\theta$. The final loss function is

$$\mathcal{L}_{\text{RL}} = -\mathbb{E}_{\hat{y} \sim \pi_\theta} \left[ \sum_{t=1}^{T} r(\hat{y}_{<t}, \mathcal{Y}_t) - \lambda \mathcal{H}(\pi_\theta(\hat{y}_{<t}, x)) \right], \tag{3}$$

where the second term inside the expectation is the negative entropy multiplied with a regularization coefficient $\lambda$. The second term encourages the exploration during training. As in (Welleck et al., 2017), we use REINFORCE (Williams, 1992) to stochastically minimize the loss function above with respect to $\pi_\theta$.

This loss function is optimal in that the return, i.e., the sum of the step-wise rewards, is maximized when both the precision and recall are maximal ($= 1$). In other words, the oracle policy, defined in Def. 2, maximizes the expected return.

However, this approach of reinforcement learning is known to be difficult, with a high variance (Peters & Schaal, 2008). This is especially true here, as the size of the state space grows exponentially with respect to the size of the target multiset, and the action space of each step is as large as the number of unique items in the class set.

## 4 EXPERIMENTS AND ANALYSIS

In this section, we extensively evaluate the proposed multiset loss function against various baseline loss functions presented throughout this paper. More specifically, we focus on its applicability and performance on image-based multiset prediction.

### 4.1 DATASETS

**MNIST Multi** MNIST Multi is a class of synthetic datasets. Each dataset consists of multiple 100x100 images, each of which contains a varying number of digits from the original MNIST (LeCun et al., 1998). We vary the size of each digit and also add clutters. In the experiments, we consider the following variants of MNIST Multi:

- **MNIST Multi (4)**: $|\mathcal{Y}| = 4$, 20-50 pixel digits
- **MNIST Multi (1-4)**: $|\mathcal{Y}| \in \{1, \ldots, 4\}$, 20-50 pixel digits
- **MNIST Multi (10)**: $|\mathcal{Y}| = 10$, 20 pixel digits

Each dataset has a training set with 70,000 examples and a test set with 10,000 examples. We randomly sample 7,000 examples from the training set to use as a validation set, and train with the remaining 63,000 examples.

**MS COCO** As a real-world dataset, we use Microsoft COCO (Lin et al., 2014) which includes natural images with multiple objects. Compared to MNIST Multi, each image in MS COCO has objects of more varying sizes and shapes, and there is a large variation in the number of object instances per image which spans from 1 to 91. The problem is made even more challenging with many overlapping and occluded objects.

To control the difficulty in order to better study the loss functions, we create the following two variants:

- **COCO Easy**: $|\mathcal{Y}| = 2$, 10,230 training examples, 24 classes

- **COCO Medium**: $|\mathcal{Y}| \in \{1, \ldots, 4\}$, 44,121 training examples, 23 classes

In both of the variants, we only include images whose $|\mathcal{Y}|$ objects are large and of common classes. An object is defined to be large if the object's area is above the 40-th percentile across the train set of MS COCO. After reducing the dataset to have $|\mathcal{Y}|$ large objects per image, we remove images containing only objects of rare classes. A class is considered rare if its frequency is less than $\frac{1}{|C|}$, where $C$ is the class set. These two stages ensure that only images with a proper number of large objects are kept. We do not use fine-grained annotation (pixel-level segmentation and bounding boxes) except for creating input-dependent rank functions from Sec. 3.2.

For each variant, we hold out a randomly sampled 15% of the training examples as a validation set. We form separate test sets by applying the same filters to the COCO validation set. The test set sizes are 5,107 for COCO Easy and 21,944 for COCO Medium.

## 4.2 MODELS

**MNIST Multi** We use three convolutional layers of channel sizes 10, 10 and 32, followed by a convolutional long short-term memory (LSTM) layer (Xingjian et al., 2015). At each step, the feature map from the convolutional LSTM layer is average-pooled spatially and fed to a softmax classifier. In the case of the one-step variant of aggregate distribution matching, the LSTM layer is skipped.

**MS COCO** We use a ResNet-34 (He et al., 2016) pretrained on ImageNet (Deng et al., 2009) as a feature extractor. The final feature map from this ResNet-34 is fed to a convolutional LSTM layer, as described for MNIST Multi above. We do not finetune the ResNet-34 based feature extractor.

In all experiments, for predicting variable-sized multisets we use the termination policy approach since it is easily applicable to all of the baselines, thus ensuring a fair comparison. Conversely, it is unclear how to extend the special class approach to the distribution matching baselines.

**Training and evaluation** For each loss, a model was trained for 200 epochs (350 for MNIST Multi 10). After each epoch, exact match was computed on the validation set. The model state with the highest validation exact match was used for evaluation on the test set.

When evaluating a trained policy, we use greedy decoding and the termination policy for determining the size of a predicted multiset. Each predicted multiset is compared against the ground-truth target multiset, and we report both the accuracy based on the exact match (EM) and F-1 score (F1), as defined in Appendix A.

Table 1: Influence of the choice of a rank function on the sequence prediction loss function

|  | **MNIST Multi (4)** | | **COCO Easy** | |
|  | EM | F1 | EM | F1 |
|---|---|---|---|---|
| **Random** | 0.920 | 0.977 | 0.721 | 0.779 |
| **Area** | 0.529 | 0.830 | 0.700 | 0.763 |
| **Spatial** | 0.917 | 0.976 | 0.675 | 0.738 |

More details about the model architectures and training are in Appendix D.

## 4.3 EXPERIMENT 1: INFLUENCE OF A RANK FUNCTION ON SEQUENCE PREDICTION

Table 3: Loss Function Comparison on the variants of MNIST Multi

| | MNIST Multi (4) | | MNIST Multi (1-4) | | MNIST Multi (10) | |
|---|---|---|---|---|---|---|
| | EM | F1 | EM | F1 | EM | F1 |
| Proposed $\mathcal{L}$ | **0.950** | **0.987** | **0.953** | **0.981** | **0.920** | **0.992** |
| $\mathcal{L}_{\text{RL}}$ | 0.912 | 0.977 | 0.945 | 0.980 | 0.665 | 0.970 |
| $\mathcal{L}_{\text{dm}}^1$ | 0.921 | 0.978 | 0.918 | 0.969 | 0.239 | 0.714 |
| $\mathcal{L}_{\text{dm}}^{\text{KL}}$ | 0.908 | 0.974 | 0.908 | 0.962 | 0.256 | 0.874 |
| $\mathcal{L}_{\text{seq}}$ | 0.906 | 0.973 | 0.891 | 0.952 | 0.592 | 0.946 |
| $\mathcal{L}_{\text{1-step}}$ | 0.210 | 0.676 | 0.055 | 0.598 | 0.032 | 0.854 |

First, we investigate the sequence loss function $\mathcal{L}_{\text{seq}}$ from Sec. 3.2, while varying a rank function. We test three alternatives: a random rank function[4] $r$ and two input-dependent rank functions $r_{\text{spatial}}$ and $r_{\text{area}}$. We compare these rank functions on MNIST Multi (4) and COCO Easy validation sets.

We present the results in Table 1. It is clear from the results that the performance of the sequence prediction loss function is dependent on the choice of a rank function. In the case of MNIST Multi, the area-based rank function was far worse than the other choices. However, this was not true on COCO Easy, where the spatial rank function was worst among the three. In both cases, we have observed that the random

Table 2: Selection Strategies

| | MNIST Multi (10) | | COCO Easy | |
|---|---|---|---|---|
| | EM | F1 | EM | F1 |
| **Greedy** | 0.920 | 0.992 | 0.702 | 0.788 |
| **Stochastic** | 0.907 | 0.990 | 0.700 | 0.790 |
| **Oracle** | 0.875 | 0.986 | 0.696 | 0.780 |

rank function performed best, and from here on, we use the random rank function in the remaining experiments. This set of experiments firmly suggests the need of an order-invariant multiset loss function, such as the multiset loss function proposed in this paper.

## 4.4 EXPERIMENT 2: EXECUTION STRATEGIES FOR THE MULTISET LOSS FUNCTION

In this set of experiments, we compare the three execution strategies for the proposed multiset loss function, illustrated in Sec. 3. They are **greedy** decoding, **stochastic** sampling and **oracle** sampling. We test them on MNIST Multi (10) and COCO Easy.

As shown in Table 2, greedy decoding and stochastic sampling, both of which consider states that are likely to be visited by the parametrized policy, outperform the oracle sampling. This is consistent with the theory by Ross et al. (2011). Although the first two strategies perform comparably to each other, across both of the datasets and the two evaluation metrics, greedy decoding tends to outperform stochastic sampling. We conjecture this is due to better matching between training and testing in the case of greedy decoding. Thus, from here on, we use greedy decoding when training a model with the proposed multiset loss function.

## 4.5 EXPERIMENT 3: LOSS FUNCTION COMPARISON

We now compare the proposed multiset loss function against the five baseline loss functions: reinforcement learning $\mathcal{L}_{\text{RL}}$, aggregate distribution matching–$\mathcal{L}_{\text{dm}}^1$ and $\mathcal{L}_{\text{dm}}^{\text{KL}}$–, its one-step variant $\mathcal{L}_{\text{1-step}}$, and sequence prediction $\mathcal{L}_{\text{seq}}$.

**MNIST Multi** We present the results on the MNIST Multi variants in Table 3. On all three variants and according to both metrics, the proposed multiset loss function outperforms all the others. The reinforcement learning based approach closely follows behind. Its performance, however, drops as the number of items in a target multiset increases. This is understandable, as the variance of policy gradient grows as the length of an episode grows. A similar behaviour was observed with sequence prediction as well as aggregate distribution matching. We were not able to train any decent models with the one-step variant of aggregate distribution matching. This was true especially in terms

---

[4]The random rank function is generated before training and held fixed. We verified that generating a new random rank function for each batch significantly decreased performance.

of exact match (EM), which we attribute to the one-step variant not being capable of modelling dependencies among the predicted items.

**MS COCO** Similarly to the results on the variants of MNIST Multi, the proposed multiset loss function matches or outperforms all the others on the two variants of MS COCO, as presented in Table 4. On COCO Easy, with only two objects to predict per example, both aggregated distribution matching (with KL divergence) and the sequence loss functions are as competitive as the proposed multiset loss. The other loss functions significantly underperform these three loss functions, as they did on MNIST Multi.

Table 4: Loss function comparison on the variants of MS COCO

|  | **COCO Easy** | | **COCO Medium** | |
|---|---|---|---|---|
|  | EM | F1 | EM | F1 |
| Proposed $\mathcal{L}$ | 0.702 | **0.788** | **0.481** | **0.639** |
| $\mathcal{L}_{\text{RL}}$ | 0.672 | 0.746 | 0.425 | 0.564 |
| $\mathcal{L}_{\text{dm}}^1$ | 0.533 | 0.614 | 0.221 | 0.085 |
| $\mathcal{L}_{\text{dm}}^{\text{KL}}$ | **0.714** | 0.763 | 0.444 | 0.591 |
| $\mathcal{L}_{\text{seq}}$ | 0.709 | 0.774 | 0.457 | 0.592 |
| $\mathcal{L}_{\text{1-step}}$ | 0.552 | 0.664 | 0.000 | 0.446 |

The performance gap between the proposed loss and the others, however, grows substantially on the more challenging COCO Medium, which has more objects per example. The proposed multiset loss outperforms the aggregated distribution matching with KL divergence by 3.7 percentage points on exact match and 4.8 on F1. This is analogous to the experiments on the MNIST Multi variants, where the performance gap increased when moving from four to ten digits.

## 4.6 ANALYSIS: ENTROPY EVOLUTION

One property of the oracle policy defined in Sec. 2.2 is that the entropy of the predictive distribution strictly decreases over time, i.e., $\mathcal{H}_{\pi_*}(y|\hat{y}_{<t}, x) > \mathcal{H}_{\pi_*}(y|\hat{y}_{\leq t}, x)$. This is a natural consequence from the fact that there is no pre-specified rank function, because the oracle policy cannot prefer any item from the others in a free label multiset. Hence, we examine here how the policy learned based on each loss function compares to the oracle policy in terms of per-step entropy. We consider the policies trained on MNIST Multi (10), where the differences among them were most clear.

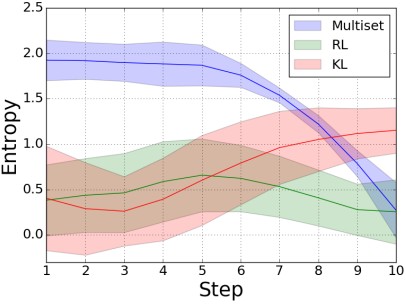

Figure 1: Comparison of per-step entropies of predictive distributions compared over the validation set.

As shown in Fig. 1, the policy trained on MNIST Multi (10) using the proposed multiset loss closely follows the oracle policy. The entropy decreases as the predictions are made. The decreases can be interpreted as concentrating probability mass on progressively smaller free labels sets. The variance is quite small, indicating that this strategy is uniformly applied for any input.

The policy trained with reinforcement learning retains a relatively low entropy across steps, with a decreasing trend in the second half. We carefully suspect the low entropy in the earlier steps is due to the greedy nature of policy gradient. The policy receives a high reward more easily by choosing one of many possible choices in an earlier step than in a later step. This effectively discourages the policy from exploring all possible trajectories during training.

On the other hand, the policy found by aggregated distribution matching ($\mathcal{L}_{\text{dm}}^{\text{KL}}$) has the opposite behaviour. The entropy in general grows as more predictions are made. To see why this is suboptimal, consider the final (10th) step. Assuming the first nine predictions $\{\hat{y}_1, ..., \hat{y}_9\}$ were correct (i.e. they form a subset of $\mathcal{Y}$), there is only one correct class left for the final prediction $\hat{y}_{10}$. The high entropy, however, indicates that the model is placing a significant amount of probability on incorrect sequences. We believe such a policy is found by minimizing the aggregated distribution matching loss function because it cannot properly distinguish between policies with increasing and decreasing entropies.

The increasing entropy also indicates that the policy has learned a rank function implicitly and is fully relying on it. Given some unknown free label multiset, inferred from the input, this policy uses the implicitly learned rank function to choose one item from this set. We conjecture this reliance on an inferred rank function, which is by definition sub-optimal,[5] resulted in lower performance of aggregate distribution matching.

## 5 CONCLUSION

We have extensively investigated the problem of multiset prediction in this paper. We rigorously defined the problem, and proposed to approach it from the perspective of sequential decision making. In doing so, an oracle policy was defined and shown to be optimal, and a new loss function, called *multiset loss*, was introduced as a means to train a parametrized policy for multiset prediction. The experiments on two families of datasets, MNIST Multi variants and MS COCO variants, have revealed the effectiveness of the proposed loss function over other loss functions including reinforcement learning, sequence, and aggregated distribution matching loss functions. The success of the proposed multiset loss brings in new opportunities of applying machine learning to various new domains, including high-energy physics.

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

## A    DEFINITIONS

We review definitions of multiset and exact match, and present multiset versions of precision, recall, and F1. For a comprehensive overview of multisets, refer to Syropoulos (2001); Singh et al. (2007).

**Multiset**    A *multiset* is a set that allows for multiple instances of elements. Multisets are unordered, i.e. $\{x, x, y\}$ and $\{x, y, x\}$ are equal. We now introduce the formal definition and convenient ways of representing a multiset.

Formally, a multiset is a pair $\mathcal{Y} = (C, \mu)$, where $C = \{c_1, ..., c_p\}$ is a *ground set*, and $\mu : C \to \mathbb{N}_{\geq 0}$ is a *multiplicity function* that maps each $c_i \in C$ to the number of times it occurs in the multiset. The multiset cardinality is defined as $|\mathcal{Y}| = \sum_{c \in C} \mu(c)$.

Notationally, a multiset can be *enumerated* by numbering each element instance and representing the multiset as a size $|\mathcal{Y}|$ set: $\mathcal{Y} = \{c_1^{(1)}, c_1^{(2)}, ..., c_1^{(\mu(c_1))}, c_2^{(1)}, ..., c_2^{(\mu(c_2))}, ..., c_p^1, ..., c_p^{(\mu(c_p))}\}$. This allows for notation such as $\sum_{c \in \mathcal{Y}}$.

An additional compact notation is $\mathcal{Y} = \{y_1, y_2, ..., y_{|\mathcal{Y}|}\}$, where each $y_i$ is an auxiliary variable referring to an underlying element $c \in C$ of the ground set.

For instance, the multiset $\mathcal{Y} = \{\text{cat}, \text{cat}, \text{dog}\}$ can be defined as $\mathcal{Y} = (C, \mu)$, where $C = \{c_1 = \text{cat}, c_2 = \text{dog}, c_3 = \text{fish}\}$, $\mu(\text{cat}) = 2, \mu(\text{dog}) = 1, \mu(\text{fish}) = 0$, and can be written as $\mathcal{Y} = \{c_1^{(1)} = \text{cat}, c_1^{(2)} = \text{cat}, c_2^{(1)} = \text{dog}\}$ or $\mathcal{Y} = \{y_1 = \text{cat}, y_2 = \text{cat}, y_3 = \text{dog}\}$.

For additional multiset representations, multiset analogues of common set operations (e.g. union, intersection, difference), and the notion of a subset, see Syropoulos (2001); Singh et al. (2007).

**Exact Match (EM)**    Two multisets *exactly match* when their elements and multiplicities are the same. For example, $\{x, y, x\}$ exactly matches $\{y, x, x\}$, while $\{x, y, x\}$ does not exactly match $\{z, y, z\}$ or $\{x, y\}$.

Formally, let $\hat{\mathcal{Y}} = (\mathcal{C}, \mu_{\hat{Y}})$, $\mathcal{Y} = (\mathcal{C}, \mu_Y)$ be multisets over a common ground set $\mathcal{C}$. Then $\hat{\mathcal{Y}}$ and $\mathcal{Y}$ exactly match if and only if $\mu_{\hat{Y}}(c) = \mu_Y(c)$ for all $c \in \mathcal{C}$. The evaluation metric $\text{EM}(\hat{\mathcal{Y}}, \mathcal{Y})$ is 1 when $\hat{\mathcal{Y}}$ and $\mathcal{Y}$ exactly match, and 0 otherwise.

Note that exact match is the same as multiset equality, i.e. $\hat{\mathcal{Y}} = \mathcal{Y}$, as defined in Singh et al. (2007).

**Precision**  Precision gives the ratio of correctly predicted elements to the number of predicted elements. Specifically, let $\hat{\mathcal{Y}} = (\mathcal{C}, \mu_{\hat{Y}})$, $\mathcal{Y} = (\mathcal{C}, \mu_Y)$ be multisets. Then

$$\text{Prec}(\hat{\mathcal{Y}}, \mathcal{Y}) = \frac{\sum_{y \in \hat{y}} I_{y \in \mathcal{Y}}}{|\hat{\mathcal{Y}}|}.$$

The summation and membership are done by enumerating the multiset. For example, the multisets $\hat{\mathcal{Y}} = \{a, a, b\}$ and $\mathcal{Y} = \{a, b\}$ are enumerated as $\hat{\mathcal{Y}} = \{a^{(1)}, a^{(2)}, b^{(1)}\}$ and $\mathcal{Y} = \{a^{(1)}, b^{(1)}\}$, respectively. Then clearly $a^{(1)} \in \mathcal{Y}$ but $a^{(2)} \notin \mathcal{Y}$.

Formally, precision can be defined as

$$\text{Prec}(\hat{\mathcal{Y}}, \mathcal{Y}) = 1 - \frac{\sum_{c \in \mathcal{C}} \max\left(\mu_{\hat{Y}}(c) - \mu_Y(c), 0\right)}{|\hat{\mathcal{Y}}|}$$

where the summation is now over the ground set $\mathcal{C}$. Intuitively, precision decreases by $\frac{1}{|\hat{\mathcal{Y}}|}$ each time an extra class label is predicted.

**Recall**  Recall gives the ratio of correctly predicted elements to the number of ground-truth elements. Recall is defined analogously to precision, as:

$$\text{Rec}(\hat{\mathcal{Y}}, \mathcal{Y}) = \frac{\sum_{y \in \hat{y}} I_{y \in \mathcal{Y}}}{|\mathcal{Y}|}.$$

Formally,

$$\text{Rec}(\hat{\mathcal{Y}}, \mathcal{Y}) = 1 - \frac{\sum_{c \in \mathcal{C}} \max\left(\mu_Y(c) - \mu_{\hat{Y}}(c), 0\right)}{|\mathcal{Y}|}.$$

Intuitively, recall decreases by $\frac{1}{|\mathcal{Y}|}$ each time an element of $\mathcal{Y}$ is not predicted.

**F1**  The F1 score is the harmonic mean of precision and recall:

$$F_1(\hat{\mathcal{Y}}, \mathcal{Y}) = 2 \cdot \frac{\text{Prec}(\hat{\mathcal{Y}}, \mathcal{Y}) \cdot \text{Rec}(\hat{\mathcal{Y}}, \mathcal{Y})}{\text{Prec}(\hat{\mathcal{Y}}, \mathcal{Y}) + \text{Rec}(\hat{\mathcal{Y}}, \mathcal{Y})}.$$

## B  PROOF OF REMARK 1

*Proof.* Note that the precision with $\hat{y}_{<t}$ is defined as

$$\text{Prec}(\hat{y}_{<t}, \mathcal{Y}) = \frac{\sum_{y \in \hat{y}_{<t}} I_{y \in \mathcal{Y}}}{|\hat{y}_{<t}|}.$$

Because $\hat{y} \sim \pi_*(\hat{y}_{<t}, x, \mathcal{Y}_t) \in \mathcal{Y}_t$,

$$\text{Prec}(\hat{y}_{\leq t}, \mathcal{Y}) = \frac{1 + \sum_{y \in \hat{y}_{<t}} I_{y \in \mathcal{Y}}}{1 + |\hat{y}_{<t}|}.$$

Then,

$$\text{Prec}(\hat{y}_{\leq t}, \mathcal{Y}) - \text{Prec}(\hat{y}_{<t}, \mathcal{Y}) = \frac{1 - \text{Prec}(\hat{y}_{<t}, \mathcal{Y})}{1 + |\hat{y}_{<t}|} \geq 0,$$

because $0 \leq \text{Prec}(\hat{y}_{<t}, \mathcal{Y}) \leq 1$ and $|\hat{y}_{<t}| \geq 0$. The equality holds when $\text{Prec}(\hat{y}_{<t}, \mathcal{Y}) = 1$.

Similarly, we start with the definition of the recall:

$$\text{Rec}(\hat{y}_{<t}, \mathcal{Y}) = \frac{\sum_{y \in \hat{y}_{<t}} I_{y \in \mathcal{Y}}}{|\mathcal{Y}|}.$$

Because $\hat{y} \sim \pi_*(\hat{y}_{<t}, x, \mathcal{Y}_t) \in \mathcal{Y}_t$,

$$\text{Rec}(\hat{y}_{\leq t}, \mathcal{Y}) = \frac{1 + \sum_{y \in \hat{y}_{<t}} I_{y \in \mathcal{y}}}{|\mathcal{Y}|}.$$

Since the denominator is identical,

$$\text{Rec}(\hat{y}_{\leq t}, \mathcal{Y}) - \text{Rec}(\hat{y}_{<t}, \mathcal{Y}) = \frac{1}{|\mathcal{Y}|} \geq 0.$$

$\square$

## C  PROOF OF REMARK 2

*Proof.* When $t = 1$,
$$\text{Prec}(\hat{y}_{\leq 1}, \mathcal{Y}) = 1,$$
because $\hat{y}_1 \sim \pi_*(\emptyset, x, \mathcal{Y}_1) \in \mathcal{Y}$. From Remark 1, we know that
$$\text{Prec}(\hat{y}_{\leq t}, \mathcal{Y}) = \text{Prec}(\hat{y}_{<t}, \mathcal{Y}),$$
when $\text{Prec}(\hat{y}_{<t}, \mathcal{Y}) = 1$. By induction, $\text{Prec}(\hat{y}_{\leq |\mathcal{Y}|}, \mathcal{Y}) = 1$.

From the proof of Remark 1, we know that the recall increases by $\frac{1}{\mathcal{y}}$ each time, and we also know that
$$\text{Rec}(\hat{y}_{\leq 1}, \mathcal{Y}) = \frac{1}{|\mathcal{Y}|},$$
when $t = 1$. After $|\mathcal{Y}| - 1$ steps of executing the oracle policy, the recall becomes

$$\text{Rec}(\hat{y}_{\leq |\mathcal{Y}|}, \mathcal{Y}) = \frac{1}{|\mathcal{Y}|} + \sum_{t'=2}^{|\mathcal{Y}|} \frac{1}{|\mathcal{Y}|} = 1.$$

$\square$

## D  MODEL DESCRIPTIONS

Figure 2: Graphical illustration of a predictor used throughout the experiments.

**Model**  An input $x$ is first processed by a tower of convolutional layers, resulting in a feature volume of size $w' \times h'$ with $d$ feature maps, i.e., $H = \phi(x) \in \mathbb{R}^{w' \times h' \times d}$. At each time step $t$, we resize the previous prediction's embedding $\text{emb}(\hat{y}_{t-1}) \in \mathbb{R}^{(w')(h')}$ to be a $w' \times h'$ tensor and concatenate it with $H$, resulting in $\tilde{H} \in \mathbb{R}^{w' \times h' \times (d+1)}$. This feature volume is then fed into a stack of convolutional LSTM layers. The output from the final convolutional LSTM layer $C \in \mathbb{R}^{w' \times h' \times q}$ is spatially average-pooled, i.e., $c = \frac{1}{w'h'} \sum_{i=1}^{w'} \sum_{j=1}^{h'} C_{i,j,\cdot} \in \mathbb{R}^q$. This feature vector $c$ is then

turned into a conditional distribution over the next item after affine transformation followed by a softmax function. When the one-step variant of aggregated distribution matching is used, we skip the convolutional LSTM layers, i.e., $c = \frac{1}{w'h'} \sum_{i=1}^{w'} \sum_{j=1}^{h'} H_{i,j,\cdot} \in \mathbb{R}^d$.

See Fig. 2 for the graphical illustration of the entire network. See Table 5 for the details of the network for each dataset.

Table 5: Network Architectures

| Data | CNN | $emb(\hat{y}_{t-1})$ | ConvLSTM |
|------|-----|------|----------|
| MNIST Multi | conv $5 \times 5$ max-pool $2 \times 2$ feat 10
conv $5 \times 5$ max-pool $2 \times 2$ feat 10
conv $5 \times 5$ max-pool $2 \times 2$ feat 32 | 81 | conv $3 \times 3$ feat 32
conv $3 \times 3$ feat 32 |
| MS COCO | ResNet-34 | 361 | conv $3 \times 3$ feat 512
conv $3 \times 3$ feat 512 |

**Preprocessing**   For MNIST Multi, we do not preprocess the input at all. In the case of MS COCO, input images are of different sizes. Each image is first resized so that its larger dimension has 600 pixels, then along its other dimension is zero-padded to 600 pixels and centered, resulting in a 600x600 image.

**Training**   The model is trained end-to-end, except ResNet-34 which remains fixed after being pretrained on ImageNet. For all the experiments, we train a neural network using Adam (Kingma & Ba, 2014) with a fixed learning rate of 0.001, $\beta$ of (0.9, 0.999) and $\epsilon$ of 1e-8. The learning rate was selected based on the validation performance during the preliminary experiments, and the other parameters are the default values. For MNIST Multi, the batch size was 64, and for COCO was 32.

**Feedforward Alternative**   While we use a recurrent model in the experiments, the multiset loss can be used with a feedforward model as follows. A key use of the recurrent hidden state is to retain the previously predicted labels, i.e. to remember the full conditioning set $\hat{y}_1, ..., \hat{y}_{t-1}$ in $p(y_t | \hat{y}_1, ..., \hat{y}_{t-1})$. Therefore, the proposed loss can be used in a feedforward model by encoding $\hat{y}_1, ..., \hat{y}_{t-1}$ in the input $x_t$, and running the feedforward model for $|\hat{\mathcal{Y}}|$ steps, where $|\hat{\mathcal{Y}}|$ is determined with a method from section 2.3. Note that compared to the recurrent model, this approach involves additional feature engineering.

