# OpenReview forum: "Loss Functions for Multiset Prediction"
_ICLR.cc/2018/Conference — Reject_

### Official Review · AnonReviewer3 · 2017-11-24
**While the proposed solution is intuitive, and can possibly be useful, the technical part of this paper is a bit weak.**

**Rating:** 5
**Confidence:** 4

**Review:**

This paper proposes a type of loss functions for the problem of multiset prediction. A detailed discussion on the intuition is provided and extensive experiments are conducted to show that this loss function indeed provides some performance gain in terms of Exact Matching and F1-score.

The idea of this paper is as follows: instead of viewing the multiset prediction task as a classification problem, this paper models it as a sequential decision problem (this idea is not new, see Welleck et al., 2017, as pointed out by the authors).  Define pi* to be the optimal policy that outputs the labels of input x in a certain way. We learn a parameterized policy pi(theta) which takes x and all previous predictions as input, and outputs a label as a new prediction. At each time step t, pi* and pi(theta) can be viewed as distributions over all remaining labels; the KL divergence is then used to calculate the difference between those two distributions. Finally, the loss function sums those KL divergences over all t. Computing this loss function directly can be intractable, so the author suggested that one can compute the entire trajectory of predictions and then do aggregation.

Admitted that such construction is quite intuitive, and can possibly be useful, the technical part of this paper seems to be rather straightforward. In particular, I think the solution to the issue of unknown T is very heuristic, making the proposed loss function less principled.


Other Detailed Comments/Issues:

-- It seems to me that models that can utilize this loss function must support varying-length inputs, e.g. LSTM. Any idea how to apply it to models with fixed-length inputs?

-- The proposed loss function assumes that T (i.e. the size of the output set) is known, which is unrealistic. To handle this, the authors simply trained an extra binary classifier that takes x and all previous predictions as the input at each time step, and decides whether or not to terminate. I think this solution is rather hacky and I’d like to see a more elegant solution, e.g. incorporate the loss on T into the proposed loss function.

-- Could you formally define “Exact Match”?

-- In Section 4.4, maybe it is better to run the stochastic sampling multiple times to reduce the variance? I would expect that the result can be quite different in different runs.

-- Any discussion on how will the classifier for T affect the experimental results?

---

> ### Author Response · Authors · 2017-12-29
> **Re: While the proposed solution is intuitive, and can possibly be useful, the technical part of this paper is a bit weak.**
>
> Thanks for your insightful review. Please find our comments below:
>
> Re: Welleck et al. [2017]
> - As noted, Welleck et al. [2017] previously proposed the sequential view of multiset prediction. We believe that the paper here contains valuable additions to the view of multiset prediction as sequential prediction. Namely, it contains a presentation of multiset prediction which is separated from a particular model architecture, comparison with existing supervised learning problems, and an extensive set of baselines that will be valuable for future work on the multiset prediction problem.
>
> Re: Issue of unknown T
> - An alternative method for predicting variable-sized multisets (i.e. the issue of unknown T) is to include an additional “END” class, similar to the token used in NLP sequence models which allows variable sentence lengths. This approach was used in Welleck et al. [2017]. In our proposed loss, this would correspond to setting the free labels set at time T+1 to be {END}.
>
> We chose to use the auxiliary stop prediction here since it was trivially applicable to all of the baseline losses, thus ensuring a fair comparison between losses. In particular, it was unclear how to extend the END class approach to the distribution matching baseline. For this reason, our experiments use the auxiliary stop prediction. We will however add a discussion of the END class approach to the manuscript (Section 2.3).
>
> Re: Models with fixed-length inputs
> - Here, the key use of the recurrent hidden state is to retain the previously predicted labels, i.e., to remember the full conditioning set \hat{y}_1,...\hat{y}_{t-1} in p(y_t|\hat{y}_1,...,\hat{y}_{t-1}). Therefore, the proposed loss can be used in a feedforward model by encoding \hat{y}_1,...,\hat{y}_{t-1}in the input x_t, and using the feedforward model for T steps. Since this involves feature engineering (e.g., to encode the variable-length sequence into a fixed-dimensional vector) and since RNNs have become a standard for sequential tasks, we use a recurrent model here. However, we appreciate your observation and have added a paragraph to Appendix C discussing this feedforward alternative.
>
> Re: extra binary classifier for T
> - Please see our comments above. In short, we will add a mention of the “END” class approach, which is a natural extension of existing approaches used in NLP sequence models. Here, we used the binary classifier to ensure the applicability to all the baselines.
>
> Re: formal definition of exact match
> - We have added a definition of exact match to Appendix A of the revised version.
>
> An intuitive way of understanding the definition is to view each multiset as a vector which counts the occurrences of each element from the class set. For example, if the possible classes are x,y,z, then the multisets A= {x,x,y}, B={y,x,x} can be represented as A=[2,1,0], B=[2,1,0]. Exact match then consists of checking whether A[i] == B[i] for all i.
>
> Re: variance across multiple runs for stochastic sampling
> - We did 9 runs of the MNIST Multi 10 experiments for each selection strategy, with a different random seed per run. The standard deviations across runs for the EM metric were 0.009, 0.01 and 0.005 for greedy, stochastic and oracle, respectively. We further tested these strategies using paired t-tests and found no significant difference between between any pair of strategies (in terms of EM.) We have just started running the same set of experiments with the COCO dataset, which we have found to be much more challenging, to ensure our observation is not limited to MNIST. As the experiments are taking much longer, we will update the result in the revision and in the response section as soon as they are completed.
>
> We however would like to note that we prefer the greedy strategy even in the case of no significant difference among these strategies due to the computational reason. The computational advantage comes from the fact that the greedy strategy does not require sampling, unlike the other ones.
>
> Re: classifier for T affecting experimental results
> - We use the same binary classification approach for each baseline. The multiset loss achieves high evaluation metrics on the variable-sized task (e.g. MNIST 1-4), which shows that the binary classifier is capable of successfully predicting T. (Otherwise, none of the baselines would have high scores on MNIST 1-4). As a result, we believe that the experimental results show the performance difference from varying the loss, given an effective binary classifier for T.

---

### Official Review · AnonReviewer2 · 2017-11-27
**multiset prediction as sequential decision making**

**Rating:** 7
**Confidence:** 3

**Review:**

This is an interesting paper, in the sense of looking at a problem such as multiset prediction in the context of sequential decision making (using a policy).

In more detail, the authors construct an oracle policy, shown to be optimal (in terms of precision and recall).  A parametrized policy instead of the oracle policy is utilized in the proposed multiset loss function, while furthermore, a termination policy facilitates the application on variable-sized multiset targets.  The authors also study other loss functions, ordered sequence prediction as well as reinforcement learning.

Results show that the proposed order-invariant loss outperforms other losses, along with a set if experiments evaluating choice of rank function for sequence prediction and selection strategies.  The experiments seem rather comprehensive, as well as the theoretical analysis.   The paper describes an interesting approach to the problem.

While the paper is comprehensive it could be improved in terms of clarity & flow (e.g., by better preparing the reader on what is to follow)

---

> ### Author Response · Authors · 2017-12-29
> **Re: multiset prediction as sequential decision making**
>
> Thank you for the comments and review. To help prepare the reader, in the updated revision we have added a description at the end of the introduction which outlines the paper structure.

---

### Official Review · AnonReviewer1 · 2017-11-29
**Technical contribution of the paper is marginal because of the lack of reliable mathematical discussion or investigation.**

**Rating:** 4
**Confidence:** 3

**Review:**

Summary:
The paper considers the prediction problem where labels are given as multisets. The authors give a definition of a loss function for multisets and show experimental results. The results show that the proposed methods optimizing the loss function perform better than other alternatives.

Comments:
The problem of predicting multisets looks challenging and interesting. The experimental results look nice. On the other hand, I have several concerns about writing and technical discussions.

First of all, the goal of the problem is not exactly stated. After I read the experimental section, I realized that the goal is to optimize the exact match score (EM) or F1 measure w.r.t. the ground truth multisets. This goal should be explicitly stated in the paper. Now then, the approach of the paper is to design surrogate loss functions to optimize these criteria.

The technical discussions for defining the proposed loss function seems not reliable for the reasons below. Therefore, I do not understand the rationale of the definition of the proposed loss function.:
- An exact definition of the term multiset is not given. If I understand it correctly, a multiset is a “set” of instances allowing duplicated ones.
- There is no definition of Prec or Rec (which look like Precision and Recall) in Remark 1. The definitions appear in Appendix, which might not be well-defined. For example, let y, Y be mutisets , y=[a, a, a] and Y = [a, b]. Then, by definition, Prec(y,Y)=3/3 =1. Is this what you meant? (Maybe, the ill-definedness  comes from the lack of definition of inclusion in a mutiset.)
- I cannot follow the proof of Remark 1 since it does not seem to take account of the randomness by the distribution \pi^*.
- I do not understand the definition of the oracle policy exactly. It seems to me that, the oracle policy knows the correct label (multi-set) \calY for each instance x and use it to construct \calY_t. But, this implicit assumption is not explicitly mentioned.
- In (1), (2) and Definition 3, what defines \calY_t? If \calY_t is determined by some “optimal” oracle, you cannot define the loss function in Def. 3 since it is not known a priori. Or, if the learner determines \calY_t, I don’t understand why the oracle policy is optimal since it depends on the learner’s choices.

Also, I expect an investigation of theoretical properties of the proposed loss function, e.g., relationship to EM or F1 or other loss functions. Without understanding the theoretical properties and the rationale, I cannot judge the goodness of the experimental results (look good though). In other words, I cannot judge the paper in a qualitative perspective, not in a quantitative view.

As a summary, I think the technical contribution of the paper is marginal because of the lack of reliable mathematical discussion or investigation.

---

> ### Author Response · Authors · 2017-12-29
> **Re: Technical contribution of the paper is marginal because of the lack of reliable mathematical discussion or investigation.**
>
> Thank you for the insightful comments. We have addressed them below and in the updated revision of the paper:
>
> Re: Problem Goal:
> - Yes, the exact match score and F1 score are used here to compare the predicted and ground truth multisets - we have added a comment when introducing multiset prediction in section 2. Moreover, minimizing the multiset loss function maximizes F1 score and exact match (due to Remarks 2 and 3, respectively); we have added a comment following Definition 3.
>
> Re: Points about technical discussion:
> - Yes, the multiset is a generalization of a set allowing multiple instances of items (here, the items are from the class set). While the bullet points at the beginning of section 2 implicitly define the notion of multiset, we will update the manuscript with an explicit definition in Appendix A of the updated revision.
>
> - Indeed, Prec and Rec are the abbreviations of Precision and Recall. Precision and recall for multisets are defined here by viewing each element of the predicted and target multisets as distinct elements (i.e. even if they are the same class), and using the precision and recall definitions in the Appendix A of the updated revision. That is, in the given example, to understand the definition of Precision / Recall it may be helpful to view y and Y as y=[a1,a2,a3], Y=[a1,b1]. Then for Precision, we have Precision(y, Y) = ½ using the definition in Appendix A.
>
> - The proof of Remark 1 does account for the randomness, as any sample from the oracle policy is guaranteed to be in the free labels set, i.e.,  \hat{y}_t \sim \pi_*(\hat{y}_t | \hat{y}_{<t},x)\in \mathcal{Y}_t with probability 1, which can be seen from Definition 2. With this in mind, could you clarify which lines of the proof are difficult to follow? We appreciate the feedback and will add comments to the proof as necessary to make the proof as clear as possible.
>
> - Indeed, the oracle policy is constructed using the target free labels multiset \mathcal{Y}_t, which relies on knowledge of \mathcal{Y}. In the updated manuscript, we have added \mathcal{Y}_t as an explicit argument of the oracle to clarify this point.
>
> - In Definition 3, \mathcal{Y}_t is defined with respect to the parametrized policy. That is, \mathcal{Y}_{t+1}=\mathcal{Y}_{t} \backslash \hat{y}_{t}, where \hat{y}_{t} \sim \pi_{\theta}. The oracle is constructed using a \mathcal{Y}_t defined with respect to its own predictions.
>
> The oracle is optimal w.r.t. to any arbitrary prefix (Remark 1). If the entire prefix was generated from the oracle, it only generates a correct multiset according to Remark 3. In other words, the oracle has the optimal behaviour given any free label set \mathcal{Y}_t.
>
> Re: Investigation of theoretical properties:
> - We have shown that the oracle policy is optimal in terms of precision and recall (and in turn F1 and exact match), so by minimizing divergence with the oracle, we can understand the loss as finding a parametrized policy whose samples are optimal. However, in this paper we have focused on empirical analysis of the proposed loss function, with positive findings. We agree with you that the convergence properties and consistency of the proposed loss function need to be theoretically investigated further in the future.
>
> Re: Goodness of experimental results:
> - The proposed loss function reduces the problem of multiset prediction into a series of supervised learning problems. Assuming that the oracle policy is included in the hypothesis space of the parametrized policy, minimizing the per-step KL divergence is a consistent estimator. However, we agree that more theoretical analysis on the proposed loss function, such as its finite-sample behavior and convergence rate, should be investigated further in the future.
>
> Our experimental results provide evidence that the parametrized policy can achieve high exact match and F1 scores after minimizing the proposed loss. The per-step entropy analysis summarized in Figure 1 provides an indirect, empirical evidence that the behavior of the parametrized policy is consistent with that of the oracle.
>
> Summary:
> We thank you for the comments, and hope that the additions of the Precision/Recall/multiset definitions, and the above clarifications about the technical analysis improve the manuscript and clarify our technical contributions.

---

### Decision · Program_Chairs · 2018-01-29
**ICLR 2018 Conference Acceptance Decision**

**Decision:**

Reject

**Comment:**

The submission addresses the problem of multiset prediction, which combines predicting which labels are present, and counting the number of each object.  Experiments are shown on a somewhat artificial MNIST setting, and a more realistic problem of the COCO dataset.

There were several concerns raised by the reviewers, both in terms of the clarity of presentation (Reviewer 1), and that the proposed solution is somewhat heuristic (Reviewer 3).  On the balance, two of three reviewers did not recommend acceptance.